# Factors Associated with Impaired Resistive Reserve Ratio and Microvascular Resistance Reserve

**DOI:** 10.3390/diagnostics13050950

**Published:** 2023-03-02

**Authors:** Tatsuro Yamazaki, Yuichi Saito, Daichi Yamashita, Hideki Kitahara, Yoshio Kobayashi

**Affiliations:** Department of Cardiovascular Medicine, Chiba University Graduate School of Medicine, Chiba 260-8670, Japan

**Keywords:** coronary microvascular dysfunction, resistance reserve ratio, microvascular resistance reserve

## Abstract

Coronary microvascular dysfunction (CMD) is described as an important subset of ischemia with no obstructive coronary artery disease. Resistive reserve ratio (RRR) and microvascular resistance reserve (MRR) have been proposed as novel physiological indices evaluating coronary microvascular dilation function. The aim of this study was to explore factors associated with impaired RRR and MRR. Coronary physiological indices were invasively evaluated in the left anterior descending coronary artery using the thermodilution method in patients suspected of CMD. CMD was defined as a coronary flow reserve <2.0 and/or index of microcirculatory resistance ≥25. Of 117 patients, 26 (24.1%) had CMD. RRR (3.1 ± 1.9 vs. 6.2 ± 3.2, *p* < 0.001) and MRR (3.4 ± 1.9 vs. 6.9 ± 3.5, *p* < 0.001) were lower in the CMD group. In the receiver operating characteristic curve analysis, RRR (area under the curve 0.84, *p* < 0.001) and MRR (area under the curve 0.85, *p* < 0.001) were both predictive of the presence of CMD. In the multivariable analysis, previous myocardial infarction, lower hemoglobin, higher brain natriuretic peptide levels, and intracoronary nicorandil were identified as factors associated with lower RRR and MRR. In conclusion, the presence of previous myocardial infarction, anemia, and heart failure was associated with impaired coronary microvascular dilation function. RRR and MRR may be useful to identify patients with CMD.

## 1. Introduction

The traditional understanding is that epicardial coronary artery disease (CAD) plays a major role in ischemic heart disease, although previous registry data showed that only less than one-half of patients suspected of angina had significant lesions in epicardial coronary arteries [1,2]. In this context, ischemia with no obstructive CAD (INOCA) has been increasingly recognized as a major etiology of ischemic heart disease [3,4], in which coronary microvascular dysfunction (CMD) and vasospastic angina are described as important subsets of INOCA in the expert consensus document [5]. Since CMD reportedly deteriorates a patient’s quality of life and prognosis [3,6], accurate identification and diagnosis are clinically relevant. Coronary flow reserve (CFR), which is the ratio of hyperemic to resting blood flow, represents coronary blood flow capacity including epicardial coronary arteries and microvasculature to accommodate an increasing demand for oxygen at excise or stress [7]. Since reduced CFR indicates the presence of CMD when no significant epicardial CAD exists, the recent European and American guidelines recommend the measurement of CFR in patients suspected of INOCA [8,9]. CFR relies on resting flow for the calculation, and, thus, hemodynamic perturbation including a change in heart rate, blood pressure, and left ventricular contractility affects CFR value [10]. Recently, resistive reserve ratio (RRR) and microvascular resistance reserve (MRR) have been proposed as novel physiological indices to represent coronary microvascular dilation function [11,12]. Given that these indices take into account the information on coronary pressure as well as flow [11,12], RRR and MRR may better estimate coronary microvascular function as compared with CFR. Indeed, previous reports showed that RRR was superior to CFR in predicting future cardiovascular events in patients with CAD [13,14]. However, data are scarce on factors related to RRR and MRR. The aim of the present study was to explore factors associated with impaired RRR and MRR.

## 2. Materials and Methods

### 2.1. Study Population

This was a retrospective, single-center study at Chiba University Hospital. Between July 2020 and June 2022, a wire-based coronary physiological assessment was conducted on 117 patients who were suspected of having CMD due to their chest pain with no apparent epicardial CAD. The invasive physiological assessment was performed in the LAD. Patients with a physiological assessment in a nonelective setting (i.e., acute coronary syndrome) (*n* = 5) and missing data (*n* = 4) were excluded. In addition, patients with angiographically significant epicardial CAD (percentage of diameter stenosis on visual assessment >50%) in the LAD were also excluded. Thus, a total of 108 patients were included in the present analysis. This study was done in accordance with the Declaration of Helsinki. The ethics committee of the Chiba University Graduate School of Medicine approved this study (Approval number: M10348, date: 27 July 2022). Informed consent was obtained in the form of opt-out.

### 2.2. Invasive Coronary Physiological Assessment

The invasive diagnostic procedure is schematized in Figure 1. After the administration of intracoronary isosorbide dinitrate, a coronary angiography was performed per local standard practice [15,16]. In the present study, wire-based invasive coronary physiological indices were measured by the bolus-saline injection thermodilution method using a 6 Fr guiding catheter with no side holes [17,18]. After equalization, the pressure sensor guidewire (PressureWire X, Abbot Vascular, Santa Clara, USA) was advanced into the distal third of the LAD, and 3 milliliters of room-temperature saline were injected into the LAD at 3 times, automatically calculating mean transit time (Tmn) with a dedicated system (CoroFlow system, Coroventis Research, Uppsala, Sweden). Simultaneously, mean aortic pressure (Pa) and distal coronary pressure (Pd) were measured. Maximum hyperemia was induced by intracoronary administration of papaverine (12 mg) or nicorandil (2 mg) [16,19]. All indices of coronary pressure and flow (i.e., Tmn) were measured at resting and hyperemic conditions.

Multiple coronary physiological indices were evaluated in this study as follows: the ratio of Pd to Pa (resting Pd/Pa), fractional flow reserve (FFR), baseline resting index (BRI), index of microcirculatory resistance (IMR), CFR, RRR, and MRR, all of which were calculated using Pa, Pd, and Tmn at rest and hyperemia. FFR was defined as Pd/Pa at hyperemia. BRI and IMR, both of which represent a coronary microvascular tone, were defined as Pd multiplied by Tmn at resting and hyperemic conditions, respectively [13,14,20,21]. CFR was defined as resting Tmn divided by hyperemic Tmn. RRR, the ratio of microvascular tone at rest to that at hyperemia was defined as follows: RRR = BRI/IMR = (resting Pd × resting Tmn)/(hyperemic Pd × hyperemic Tmn) = CFR × (resting Pd/hyperemic Pd) [13,14]. In the present study, MRR was calculated by using indices obtained by a bolus-saline thermodilution method, rather than measured by absolute coronary blood flow using a continuous-saline thermodilution method. The definition of MRR was as follows: MRR = CFR × (resting Pa/hyperemic Pd) = (CFR/FFR) × (resting Pa/hyperemic Pa) = RRR × (resting Pa/resting Pd) [12,22]. The cut-off values for abnormal FFR, IMR, and CFR were determined as ≤0.80, ≥25, and <2.0, respectively [8,9]. In the present study, patients with abnormal CFR and/or IMR (i.e., CFR < 2.0 and/or IMR ≥ 25) were defined as having CMD [5,8,9].

### 2.3. Endpoints and Statistical Analysis

The primary interest of this study was to explore factors associated with impaired (i.e., lower) RRR and MRR. All statistical analyses were performed using JMP pro version 16.0 (SAS Institute Inc., Cary, CA, USA). Continuous variables were expressed as mean ± standard deviation and compared with the Student t-test. Categorical variables were expressed as frequency (%) and assessed with Fisher’s exact test. The normal distribution was visually evaluated with histograms. Due to the skewed distribution, a log transformation was performed to assess the level of brain natriuretic peptide (BNP). The receiver operating characteristic (ROC) curve analyses were performed to assess the best cut-off value of RRR and MRR for predicting CMD. Univariable and multivariable linear regression analyses were performed to explore factors related to coronary physiological indices. In the regression models, we included variables reportedly affecting coronary physiological statuses such as age, sex, body mass index, diabetes, hypertension, previous myocardial infarction (MI), renal function assessed with estimated glomerular filtration rate, anemia evaluated with a hemoglobin level, heart failure estimated by log-transformed BNP, and hyperemic agent (i.e., intracoronary papaverine versus nicorandil) [23,24,25,26,27,28,29,30,31,32]. The results of the regression analysis are displayed in a heat map. As a sensitivity analysis, the univariable and multivariable linear regression analyses were performed after excluding cases in which intracoronary nicorandil was used to achieve maximum hyperemia. A value of *p* < 0.05 was considered statistically significant. No corrections for multiple comparisons were performed.

## 3. Results

Of the 108 patients, 26 (24.1%) had CMD (Table 1). Baseline characteristics between patients with and without CMD are summarized in Table 1. Patients with CMD were more likely to be women, while other characteristics were similar between the two groups (Table 1).

Coronary physiological findings are shown in Table 2. To archive maximum hyperemia, intracoronary papaverine, and nicorandil were used in 62.0% and 38.0%, respectively. The use of nicorandil was more frequent in women than in men (79.3% vs. 22.8%, *p* < 0.001). FFR, BRI, and IMR were significantly higher and CFR, RRR, and MRR were lower in patients with CMD than those without (Table 2).

The ROC curve analyses showed that RRR and MRR were both predictive of the presence of CMD (Figure 2). With the best cut-off value, the sensitivity, specificity, positive and negative predictive values, and diagnostic accuracy of RRR ≤ 3.4 and MRR ≤ 3.7 for CMD were 0.77, 0.84, 0.61, 0.92, and 0.82, and 0.77, 0.87, 0.65, 0.92, and 0.84, respectively.

In the univariable analysis, female gender, the presence of previous MI, a lower hemoglobin level, higher log-transformed BNP, and intracoronary nicorandil as a hyperemic agent were significantly associated with lower RRR and MRR (Figure 3).

Multivariable analysis indicated previous MI, a lower hemoglobin level, higher log-transformed BNP, and intracoronary nicorandil as predictors of lower RRR and MRR (Figure 4).

When excluding cases in which intracoronary nicorandil was used to achieve maximum hyperemia (Table 3 and Table 4), the overall results were similar (Figure 5, Figure 6 and Figure 7).

## 4. Discussion

The present study demonstrated that among patients suspected of CMD, approximately one quarter had invasively assessed CFR <2.0 and/or IMR ≥25. Patients with CMD had a lower CFR, RRR, and MRR than those without. Multivariable analysis identified previous MI, anemia, and heart failure as factors associated with impaired RRR and MRR. To our knowledge, this is the first report exploring predictors of the novel indices, RRR and MRR, for evaluating coronary microvascular dilation function.

### 4.1. RRR and MRR

Recently, INOCA has been of clinical interest, in which CMD is the main subset [5]. Since invasive identification and subsequent medical therapy were shown to improve the quality of life in patients with INOCA [33,34], an accurate diagnosis is clinically relevant. Although CFR (<2.0) and IMR (≥25) are suggested to define INOCA in the guidelines [8,9], whether the two indices can accurately identify patients with CMD remains uncertain. CFR is affected by hemodynamic perturbation such as a change in heart rate, blood pressure, and left ventricular contractility [10], and IMR is influenced by the amount of myocardium subtended to the location of the pressure-temperature sensor [35]. To overcome the limitations of CFR and IMR measurement, recently emerged RRR and MRR may be useful for evaluating coronary microvascular dilation function. While previous studies showed that only one physiologic index, including FFR, CFR, or IMR, was unable to fully discriminate patients at higher risks of clinical events, RRR is an integrated physiologic index of both coronary flow and pressure, potentially resulting in better risk stratification in CMD [13,14]. In fact, a previous single-center study (*n* = 1692) showed that RRR (mean value 2.88) was useful to stratify risks for all-cause mortality in patients with angina or ischemia and nonobstructive CAD, with the best cut-off value of 2.62 [14]. Another patient-level pooled cohort in Korea, Japan, and Spain demonstrated that lower RRR was associated with worse clinical outcomes in a stepwise manner and that even in patients with preserved FFR (>0.80) and CFR (>2.0), lower RRR (<3.5) was related to an increased risk of patient-oriented composite outcomes during the long-term follow-up [13]. The cut-off (median) value for predicting outcomes suggested in the pooled data (i.e., 3.5) was in line with that for the presence of CMD in the present study (i.e., 3.4), although RRR in the present study was numerically higher than that of previous studies [13,14]. In the prior pooled data, >30% of patients had CFR <2.0 [13], whereas approximately 10% did in the present study, suggesting that our study cohort represented relatively preserved coronary microvascular function.

MRR was originally developed as the index measured by absolute coronary blood flow with a continuous-saline thermodilution method using a dedicated microcatheter [36]. MRR is conceptually specific for microcirculation and independent of myocardial mass [12]. Although MRR was calculated by using indices obtained with a bolus-saline thermodilution method in the present study, it has the potential to avoid influence with epicardial CAD and the amount of myocardium [12]. The suggested cut-off value of MRR for the presence of CMD in this study (i.e., 3.7) was slightly higher than that of RRR, which may be reasonable due to the calculation formula (i.e., MRR = RRR × [resting Pa/resting Pd]) [12,22]. Given that CFR, RRR, and MRR were all significantly lower in patients with CMD than those without, multiple physiological assessments can aid in identifying patients with CMD. Further studies are needed to elucidate the cut-off values of RRR and MRR and whether the novel indices are superior to conventional invasive indices such as CFR and IMR in estimating coronary microvascular function.

### 4.2. Factors Associated with RRR and MRR

It is conceivable that CMD, greater resting coronary blood flow, or both result in impaired microvascular dilation response (i.e., RRR and MRR) [13,14], which are reportedly associated with several clinical and procedural factors. For instance, FFR was preserved while IMR, CFR, RRR, and MRR were more impaired in women than in men in the univariable analysis in the present study, probably due to the longer hyperemic Tmn (slower coronary blood flow) (Figure 3). However, previous studies showed that women had lower CFR, with a shorter resting Tmn (faster coronary blood flow) [37,38]. The longer hyperemic Tmn in women may be confounded with the higher likelihood of nicorandil use as a hyperemic agent. Indeed, when excluding cases in which maximum hyperemia was achieved by intracoronary nicorandil, the female gender was no longer associated with lower CFR, RRR, and MRR in both univariable and multivariable analyses (Figure 6 and Figure 7). Women are likely to have impaired CFR, though the underlying mechanisms remain unclear. Additionally, a previous study in which prognostic implications of RRR were evaluated in patients with INOCA showed that the rate of women was higher in patients with reduced RRR (<2.62) than in their counterparts [14]. In the multivariable adjustment with hemoglobin and BNP levels, female gender was no longer a significant factor associated with CFR, RRR, and MRR in the present study, suggesting that anemia may play a role in a higher likelihood of CMD in women. Apart from gender differences, several patient characteristics such as older age and the presence of diabetes are known to be associated with impaired CFR [24,39]. A recent retrospective study showed that MRR was significantly lower in diabetic patients with suspected angina and nonobstructive CAD than those without diabetes [31], and diabetes was also reportedly associated with lower RRR [13,14]. Although the present study did not show the direct relation of diabetes to CFR, the multivariable analysis indicated that patients with diabetes had nonsignificantly lower RRR and MRR.

In this study, a multivariable analysis identified previous MI, anemia, and heart failure as factors associated with impaired RRR and MRR. In a recent prospective study in which invasive measures of coronary microvascular function such as CFR and IMR were repeatedly evaluated in patients undergoing primary percutaneous coronary artery intervention for ST-segment-elevation MI, IMR remained high (i.e., 25.6 ± 17.8) at one month after the index event [40]. Patients with a history of MI are likely to have coronary arteriosclerosis and impaired microvascular function [41], probably resulting in lower RRR and MRR. The increased resting and impaired hyperemic coronary blood flow in patients with anemia and heart failure were reported in previous investigations, as shown in the present study [42,43,44], supported by the fact that lower hemoglobin and higher BNP levels were associated with a shorter resting Tmn and BRI in the univariable models (Figure 3). It is conceivable that the increased resting coronary blood flow reflected a patient condition where hyperemic status, at least partially, was achieved even at rest, preventing “additional” maximum hyperemia by intracoronary administration of papaverine and nicorandil. Although intracoronary nicorandil is safe and effective to induce hyperemia [19], an achievable hyperemic effect by intracoronary papaverine may be greater as compared with nicorandil [32], leading to the significant influence of different hyperemic agents (i.e., papaverine vs. nicorandil) on RRR and MRR. In previous reports, a hyperemic effect of intracoronary papaverine is induced earlier and lasts longer than that of nicorandil [45,46]. However, when excluding cases in which nicorandil was used for inducing maximum hyperemia, the overall results were similar. Thus, we believe that the presence of previous MI, anemia, and heart failure may be significant predictors of impaired RRR and MRR. To estimate whether a patient has CMD in clinical practice, these factors may be taken into account.

### 4.3. Study Limitations

There were some limitations in the present study. This was a retrospective, single-center, observational study, and the sample size was modest. The number of patients included in this study may be acceptable to perform the multivariable analyses [47], however, a larger sample size would be preferred. Although the present study included patients suspected of CMD, only one quarter had CFR <2.0 and/or IMR ≥25. Noninvasive stress tests to evaluate myocardial ischemia were not performed in a uniform manner and thus, the data were not available. Different hyperemic agents, such as intracoronary papaverine, nicorandil, intravenous adenosine, and adenosine triphosphate reportedly have different characteristics in safety, efficacy, and availability in real-world clinical practice. The decision of physiological measurement and the selection of hyperemic agent were left to the operator′s discretion. Even though the sensitivity analysis confirmed similar results between the entire study population and cases in which intracoronary papaverine was used to achieve maximum hyperemia, a selection bias is possible. In this study, we estimated MRR by using a bolus-saline thermodilution method rather than using a continuous-saline thermodilution method as done in previous reports [12,22].

## 5. Conclusions

Coronary microvascular dilation function assessed with RRR and MRR was impaired in patients with CMD, both of which may help estimate coronary microvascular function. The presence of previous MI, anemia, and heart failure were identified as factors associated with lower RRR and MRR. The clinical usefulness of RRR and MRR beyond conventional physiological indices such as CFR and IMR deserves further investigation.

## Figures and Tables

**Figure 1 diagnostics-13-00950-f001:**
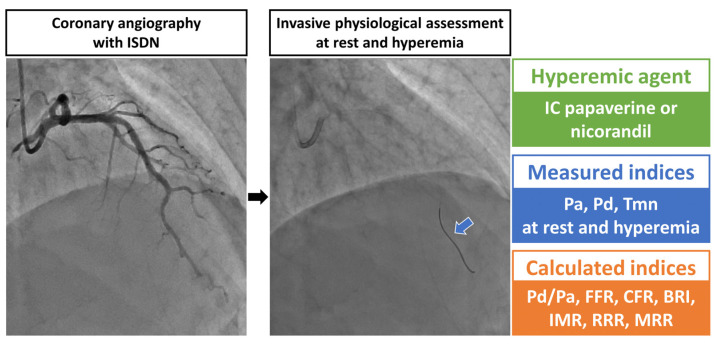
After administration of ISDN, coronary angiography was performed. Subsequently, the pressure sensor guidewire (PressureWire X, Abbot Vascular, Santa Clara, CA, USA) was advanced into the distal third of the left anterior descending artery after equalization. Three milliliters of room-temperature saline were injected into the left anterior descending artery at 3 times at rest, automatically calculating Tmn. Mean Pa and Pd were measured simultaneously. Tmn, Pa, and Pd were again measured under maximum hyperemia using intracoronary papaverine or nicorandil. Pd/Pa at rest, FFR, CFR, BRI, IMR, RRR, and MRR were calculated. A blue arrow indicates the pressure sensor guidewire. BRI, baseline resistance index; CFR, coronary flow reserve; FFR, fractional flow reserve; IC, intracoronary; IMR, index of microcirculatory resistance; ISDN, intracoronary isosorbide dinitrate; MRR, microvascular resistance reserve; Pa, mean aortic pressure; Pd, mean distal coronary pressure; RRR, resistance reserve ratio; Tmn, mean transit time.

**Figure 2 diagnostics-13-00950-f002:**
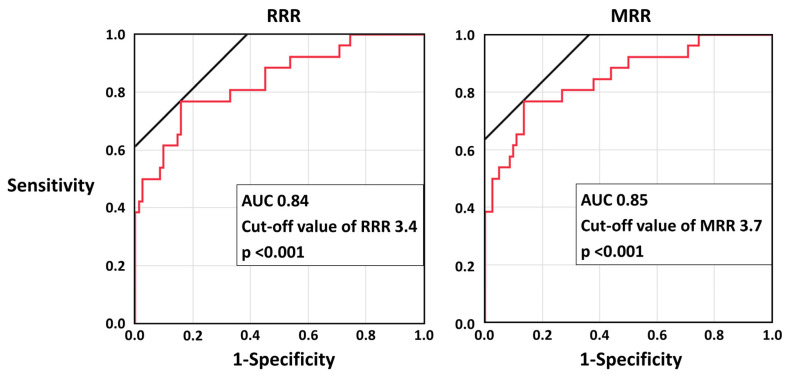
The receiver operating characteristic curve analyses of RRR and MRR for CMD. AUC, area under the curve; CMD, coronary microvascular dysfunction; MRR, microvascular resistance reserve, RRR; resistance reserve ratio.

**Figure 3 diagnostics-13-00950-f003:**
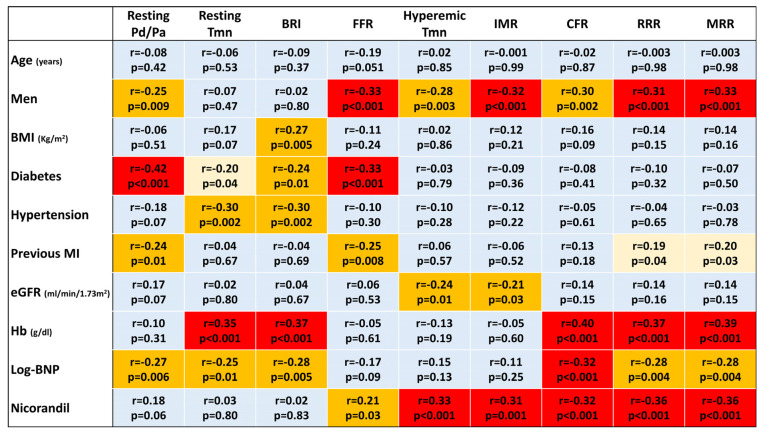
A heat map for univariable linear regression analyses. A cell in light blue indicates no significant association (*p* ≥ 0.05). When significant associations are found (*p* < 0.05), cells are displayed in light (−0.20 ≤ *r* ≤ 0.20) or warm yellow (−0.30 ≤ *r* < −0.20 or 0.20 < *r* ≤ 0.30) or red (*r* < −0.30 or *r* > 0.30). BMI, body mass index; BNP, brain natriuretic peptide; BRI, baseline resistance index; CFR, coronary flow reserve; eGFR, estimated glomerular filtration rate; FFR, fractional flow reserve; IMR, index of microcirculatory resistance; MI, myocardial infarction; MRR, microvascular resistance reserve; Pd/Pa, ratio of distal coronary pressure to aortic pressure; RRR, resistive reserve ratio; Tmn, mean transit time.

**Figure 4 diagnostics-13-00950-f004:**
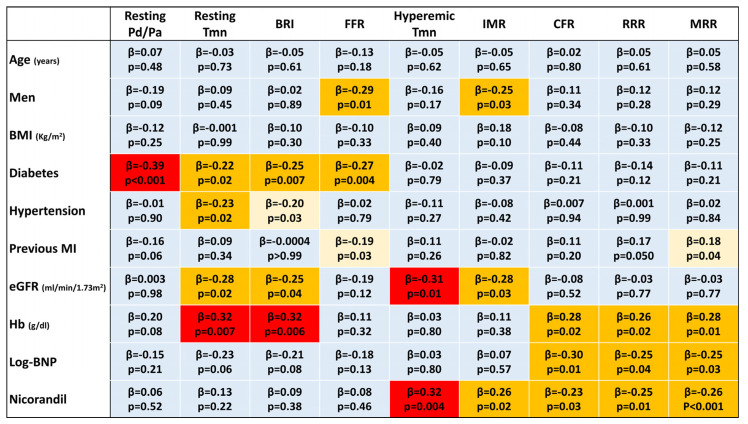
A heat map for multivariable linear regression analyses. A cell in light blue indicates no significant association (*p* ≥ 0.05). When significant associations are found (*p* < 0.05), cells are displayed in light (−0.20 ≤ *β* ≤ 0.20) or warm yellow (−0.30 ≤ *β* < −0.20 or 0.20 < *β* ≤ 0.30) or red (*β* < −0.30 or *β* > 0.30). BMI, body mass index; BNP, brain natriuretic peptide; BRI, baseline resistance index; CFR, coronary flow reserve; eGFR, estimated glomerular filtration rate; FFR, fractional flow reserve; IMR, index of microcirculatory resistance; MI, myocardial infarction; MRR, microvascular resistance reserve; Pd/Pa, ratio of distal coronary pressure to aortic pressure; RRR, resistive reserve ratio; Tmn, mean transit time.

**Figure 5 diagnostics-13-00950-f005:**
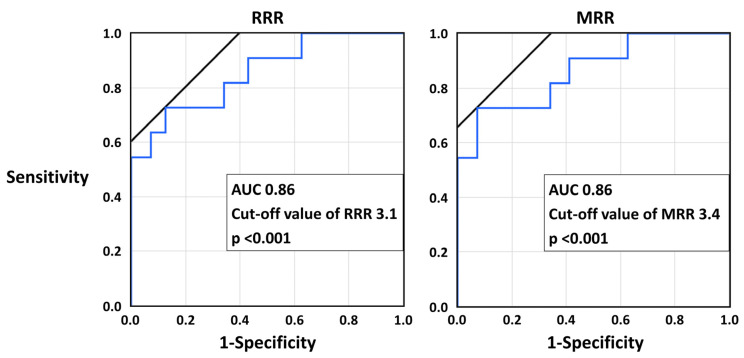
The receiver operating characteristic curve analyses of RRR and MRR for CMD in cases in which papaverine was used as a hyperemic agent. AUC, area under the curve; CMD, coronary microvascular dysfunction; MRR, microvascular resistance reserve, RRR; resistance reserve ratio.

**Figure 6 diagnostics-13-00950-f006:**
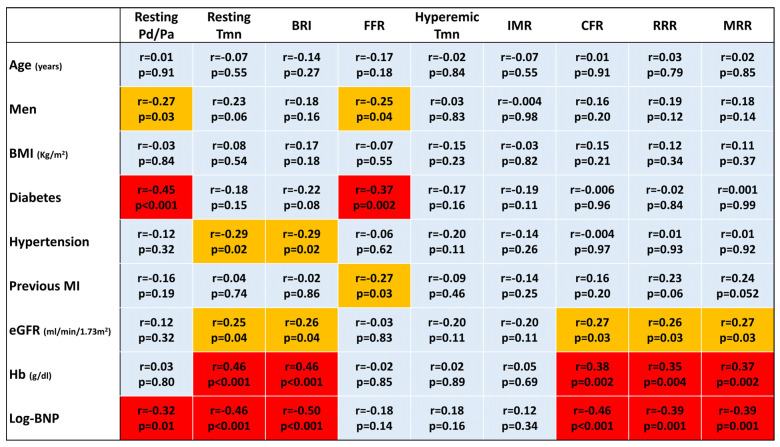
A heat map for univariable linear regression analyses in cases in which papaverine was used as a hyperemic agent. A cell in light blue indicates no significant association (*p* ≥ 0.05). When significant associations are found (*p* < 0.05), cells are displayed in light (−0.20 ≤ *r* ≤ 0.20) or warm yellow (−0.30 ≤ *r* < −0.20 or 0.20< *r* ≤ 0.30) or red (*r* < −0.30 or *r* > 0.30). BMI, body mass index; BNP, brain natriuretic peptide; BRI, baseline resistance index; CFR, coronary flow reserve; eGFR, estimated glomerular filtration rate; FFR, fractional flow reserve; IMR, index of microcirculatory resistance; MI, myocardial infarction; MRR, microvascular resistance reserve; Pd/Pa, ratio of distal coronary pressure to aortic pressure; RRR, resistive reserve ratio; Tmn, mean transit time.

**Figure 7 diagnostics-13-00950-f007:**
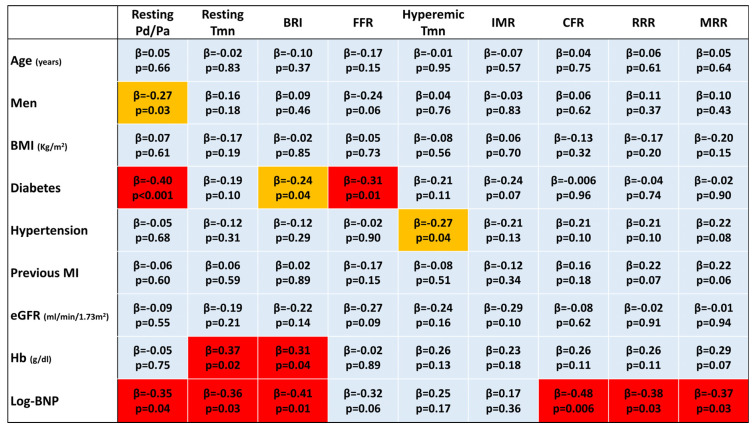
A heat map for multivariable linear regression analyses in cases in which papaverine was used as a hyperemic agent. A cell in light blue indicates no significant association (*p* ≥ 0.05). When significant associations are found (*p* < 0.05), cells are displayed in light (−0.20 ≤ *β* ≤ 0.20) or warm yellow (−0.30 ≤ *β* < −0.20 or 0.20 < *β* ≤ 0.30) or red (*β* <−0.30 or *β* > 0.30). BMI, body mass index; BNP, brain natriuretic peptide; BRI, baseline resistance index; CFR, coronary flow reserve; eGFR, estimated glomerular filtration rate; FFR, fractional flow reserve; IMR, index of microcirculatory resistance; MI, myocardial infarction; MRR, microvascular resistance reserve; Pd/Pa, ratio of distal coronary pressure to aortic pressure; RRR, resistive reserve ratio; Tmn, mean transit time.

**Table 1 diagnostics-13-00950-t001:** Baseline characteristics.

Variable	All (*n* = 108)	CMD (*n* = 26)	Non-CMD (*n* = 82)	*p* Value
Age (years)	68.5 ± 11.8	67.3 ± 13.0	68.9 ± 11.5	0.56
Men	79 (73.1%)	14 (53.9%)	65 (79.3%)	0.02
Body mass index (kg/m^2^)	24.4 ± 3.6	24.6 ± 4.3	24.4 ± 3.4	0.78
Hypertension	74 (68.5%)	18 (69.2%)	56 (68.3%)	1.00
Diabetes	31 (28.7%)	8 (30.8%)	23 (28.1%)	0.81
Dyslipidemia	85 (78.7%)	19 (73.1%)	66 (80.5%)	0.42
Current smoking	14 (13.0%)	4 (15.4%)	10 (12.2%)	0.74
Chronic kidney disease	37 (34.3%)	13 (50.0%)	24 (29.3%)	0.06
Hemodialysis	7 (6.5%)	4 (15.4%)	3 (3.7%)	0.06
Previous MI	12 (11.1%)	2 (7.7%)	10 (12.2%)	0.73
Atrial fibrillation	14 (13.0%)	1 (3.9%)	13 (15.9%)	0.18
Hemoglobin (g/dL)	13.7 ± 1.9	13.1 ± 2.1	13.9 ± 1.9	0.09
eGFR (mL/min/1.73 m^2^)	65.5 ± 23.5	59.1 ± 29.3	67.6 ± 21.2	0.11
LDL cholesterol (mg/dL)	96 ± 31	90 ± 23	98 ± 33	0.29
Glycated hemoglobin (%)	6.0 ± 0.7	6.3 ± 1.0	6.0 ± 0.6	0.06
Log-BNP	3.47 ± 1.35	3.89 ± 1.50	3.34 ± 1.23	0.08
Medical treatment				
Antiplatelet agent	59 (54.6%)	13 (50.0%)	46 (56.1%)	0.65
Statin	69 (63.9%)	18 (69.2%)	51 (62.2%)	0.64
β-blocker	37 (34.3%)	9 (34.6%)	28 (34.2%)	1.00
ACE-i or ARB	46 (42.6%)	10 (38.5%)	36 (43.9%)	0.66
Calcium channel blocker	57 (52.8%)	15 (57.7%)	42 (51.2%)	0.65
Nitrate	18 (16.7%)	7 (26.9%)	11 (13.4%)	0.13

ACE-i, angiotensin-converting enzyme inhibitor; ARB, angiotensin II receptor blocker; BNP, brain natriuretic peptide; CMD, coronary microvascular dysfunction; eGFR, estimated glomerular filtration rate; LDL, low-density lipoprotein; MI, myocardial infarction.

**Table 2 diagnostics-13-00950-t002:** Physiological findings.

Variable	All (*n* = 108)	CMD (*n* = 26)	Non-CMD (*n* = 82)	*p* Value
Hyperemic agent				
Papaverine	67 (62.0%)	11 (42.3%)	56 (68.3%)	0.02
Nicorandil	41 (38.0%)	15 (57.7%)	26 (31.7%)	
Physiologic findings				
Resting Pd/Pa	0.93 ± 0.03	0.93 ± 0.04	0.93 ± 0.02	0.75
FFR	0.86 ± 0.06	0.89 ± 0.06	0.86 ± 0.06	0.04
FFR ≤ 0.80	19 (17.6%)	3 (11.5%)	16 (19.5%)	0.55
Resting Tmn	0.94 ± 0.49	1.06 ± 0.67	0.91 ± 0.41	0.17
Hyperemic Tmn	0.24 ± 0.15	0.40 ± 0.19	0.19 ± 0.08	<0.001
BRI	80.4 ± 43.3	99.0 ± 63.3	74.5 ± 33.0	0.01
IMR	17.7 ± 11.5	31.5 ± 14.4	13.3 ± 5.5	<0.001
IMR ≥25	20 (18.5%)	20 (76.9%)	0 (0.0%)	<0.001
CFR	4.7 ± 2.6	2.8 ± 1.7	5.3 ± 2.5	<0.001
CFR <2.0	12 (11.1%)	12 (46.2%)	0 (0.0%)	<0.001
RRR	5.5 ± 3.2	3.1 ± 1.9	6.2 ± 3.2	<0.001
MRR	6.1 ± 3.5	3.4 ± 1.9	6.9 ± 3.5	<0.001

BRI, baseline resistance index; CFR, coronary flow reserve; CMD, coronary microvascular dysfunction; FFR, fractional flow reserve; IMR, index of microcirculatory resistance; MRR, microvascular resistance reserve; Pd/Pa, the ratio of distal coronary pressure to aortic pressure; RRR, resistive reserve ratio; Tmn, mean transit time.

**Table 3 diagnostics-13-00950-t003:** Baseline characteristics in patients with physiological testing using papaverine.

Variable	All (*n* = 67)	CMD (*n* = 11)	Non-CMD (*n* = 56)	*p* Value
Age (years)	68.6 ± 11.6	67.3 ± 13.3	68.8 ± 11.3	0.68
Men	61 (91.0%)	10 (90.9%)	51 (91.1%)	1.00
Body mass index (kg/m^2^)	24.9 ± 3.5	25.3 ± 3.8	24.8 ± 3.4	0.69
Hypertension	49 (73.1%)	8 (72.7%)	41 (73.2%)	1.00
Diabetes	19 (28.4%)	3 (27.3%)	16 (28.6%)	1.00
Dyslipidemia	52 (77.6%)	7 (63.6%)	45 (80.4%)	0.25
Current smoking	12 (17.9%)	3 (27.3%)	9 (16.1%)	0.40
Chronic kidney disease	21 (31.3%)	7 (63.6%)	14 (25.0%)	0.03
Hemodialysis	6 (9.0%)	3 (27.3%)	3 (5.4%)	0.051
Previous MI	10 (14.9%)	0 (0.0%)	10 (17.9%)	0.19
Atrial fibrillation	10 (14.9%)	0 (0.0%)	10 (17.9%)	0.19
Hemoglobin (g/dL)	14.0 ± 2.0	13.4 ± 2.6	14.1 ± 1.9	0.30
eGFR (mL/min/1.73 m^2^)	63.9 ± 24.4	45.1 ± 31.3	67.6 ± 21.2	0.004
LDL cholesterol (mg/dL)	92 ± 29	87 ± 19	93 ± 30	0.48
Glycated hemoglobin (%)	6.0 ± 0.6	6.1 ± 0.5	6.0 ± 0.6	0.34
Log-BNP	3.52 ± 1.47	4.49 ± 1.79	3.32 ± 1.33	0.01
Medical treatment				
Antiplatelet	39 (58.2%)	6 (54.6%)	33 (58.9%)	1.00
Statin	40 (59.7%)	7 (63.6%)	33 (58.9%)	1.00
β-blocker	23 (34.3%)	4 (36.4%)	19 (33.9%)	1.00
ACE-i or ARB	28 (41.8%)	5 (45.5%)	23 (41.1%)	1.00
Calcium channel blocker	36 (53.7%)	6 (54.6%)	30 (53.6%)	1.00
Nitrate	10 (14.9%)	3 (27.3%)	7 (12.5%)	0.35

ACE-i, angiotensin-converting enzyme inhibitor; ARB, angiotensin II receptor blocker; BNP, brain natriuretic peptide; CMD, coronary microvascular dysfunction; eGFR, estimated glomerular filtration rate; LDL, low-density lipoprotein; MI, myocardial infarction.

**Table 4 diagnostics-13-00950-t004:** Physiological findings under intracoronary papaverine-induced hyperemia.

Variable	All (*n* = 67)	CMD (*n* = 11)	Non-CMD (*n* = 56)	*p* Value
Resting Pd/Pa	0.93 ± 0.03	0.92 ± 0.03	0.93 ± 0.02	0.17
FFR	0.85 ± 0.07	0.87 ± 0.06	0.85 ± 0.07	0.49
FFR ≤0.80	16 (23.9%)	3 (27.3%)	13 (23.2%)	0.72
Resting Tmn	0.93 ± 0.49	0.96 ± 0.75	0.93 ± 0.42	0.86
Hyperemic Tmn	0.20 ± 0.11	0.34 ± 0.18	0.18 ± 0.07	<0.001
BRI	79.7 ± 41.9	92.4 ± 74.0	77.2 ± 32.7	0.28
IMR	14.9 ± 9.9	28.0 ± 16.5	12.3 ± 5.1	<0.001
IMR ≥25	8 (11.9%)	8 (72.7%)	0 (0.0%)	<0.001
CFR	5.4 ± 2.9	2.8 ± 2.0	5.9 ± 2.8	<0.001
CFR <2.0	6 (9.0%)	6 (54.6%)	0 (0.0%)	<0.001
RRR	6.4 ± 3.6	3.2 ± 2.3	7.0 ± 3.4	<0.001
MRR	7.1 ± 3.9	3.5 ± 2.4	7.8 ± 3.7	<0.001

BRI, baseline resistance index; CFR, coronary flow reserve; CMD, coronary microvascular dysfunction; FFR, fractional flow reserve; IMR, index of microcirculatory resistance; MRR, microvascular resistance reserve; Pd/Pa, ratio of distal coronary pressure to aortic pressure; RRR, resistive reserve ratio; Tmn, mean transit time.

## Data Availability

The data of this study are available upon reasonable request.

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
