# Peer review of "Factors Associated with Impaired Resistive Reserve Ratio and Microvascular Resistance Reserve"

_diagnostics, 2023, doi:10.3390/diagnostics13050950_

Round 1

Reviewer 1 Report

In the manuscript by Yamazaki et all, coronary microvascular dysfunction is examined in more details. It is a retrospective study of 117 patients undergoing invasive intracoronary hemodynamic assessment.

CMD was defined as coronary flow reserve <2.0 and/or index of microcirculatory resistance>25. Univariable and multivariable linear regression was performed to identify key factors associated with lower RRR and MRR. RRR and MRR are relatively new parameters used to describe the coronary microcirculation and potentially they are more accurate than coronary flow reserve which depends on parameters like blood pressure and heart rate. The coronary microcirculation is an important clinical entity which however is hard to quantify. It has gained increased focus in the recent years as a course for chest pain without obstructive coronary artery disease.

The study is well-written but suffers from limitations due to its retrospective nature and the relatively small number of patients.

Retrospective study with different vasodilators. How does it influence outcome?

Multivariable linear regression is mainly considered meaningful having when the number of subjects exceeds 20 x number of variables. This is not fulfilled in most of the analysis.

Minor

A figure illustrating the measuring catheter and the different hemodynamic parameters (measured and derived) could be helpful.

Page 2, line 99: ‘contentious’ should be ‘continuous’

Statistics: Student t-test was used. Did you examine for normality? Otherwise, non-parametric statistics would be more appropriate.

Bolus-saline thermodilution was used instead of continuous-saline thermodilution. What difference is this expected to make?

Reviewer 2 Report

I believe that this paper deserves to be reported because it uses RRR and MRR as new indicators of coronary microvascular dysfunction (CMD) and clarifies the factors that influence them. The authors should comment on the following points

#1 The authors performed a separate analysis regarding each loading drug, such as papaverine, which is further described in the study limitation, but I think it is important to note that two intracoronary loading drugs were used in this study. The authors should indicate how much the time to increase blood flow differs with these drugs when administered intracoronary. In addition, the authors should comment again that in this study, the frequency of use of loading agents differs by gender, hemodialysis patients, patients with previous myocardial infarction, etc. Finally, the authors had better state which drug the authors believe is preferable, or whether intravenously administered drugs are more preferable, etc. These can be described in the "Study limitations".

#2 Although patients with previous myocardial infarction are mentioned in this study, are other patients with cardiomyopathy or other heart failure included? 
